# Temporal and Spatial Variations of Secchi Depth and Diffuse Attenuation Coefficient from Sentinel-2 MSI over a Large Reservoir

**Gonçalo Rodrigues** [1,*]📵, **Miguel Potes** [1]📵, **Maria João Costa** [1,2]📵, **Maria Helena Novais** [1,3,4]📵, **Alexandra Marchã Penha** [1,3], **Rui Salgado** [1,2]📵 **and Maria Manuela Morais** [1,3,5]

1   Institute of Earth Sciences, Institute for Advanced Studies and Research, University of Évora, 7000-671 Évora, Portugal; mpotes@uevora.pt (M.P.); mjcosta@uevora.pt (M.J.C.); hnovais@uevora.pt (M.H.N.); mapenha@uevora.pt (A.M.P.); rsal@uevora.pt (R.S.); mmorais@uevora.pt (M.M.M.)
2   Department of Physics, School of Science and Technology, University of Évora, 7000-671 Évora, Portugal
3   Water Laboratory, University of Évora, P.I.T.E. Rua da Barba Rala No. 1, 7005-345 Évora, Portugal
4   Renewable Energies Chair, University of Évora, Casa Cordovil, Rua D. Augusto Eduardo Nunes, n.7, 7000-651 Évora, Portugal
5   Department of Biology, School of Science and Technology, University of Évora, 7000-671 Évora, Portugal
*   Correspondence: grodrigues@uevora.pt

**Abstract:** The Alqueva reservoir (South of Portugal) in the Guadiana river basin constitutes the most important water resource in southern Portugal for domestic and agricultural consumption. We present a methodology developed to characterize spatial and temporal variations of Secchi depth and diffuse attenuation coefficient (both related to dissolved/suspended particles and to water transparency), using high spatial resolution satellite images from Sentinel-2 Multi-Spectral Instrument (MSI). Empirical relations between satellite retrievals of surface reflectances and in situ measurements of water parameters were defined and applied to the entire reservoir for spatial and temporal analysis in the period July 2017–June 2019, useful in the identification of microalgae blooms and rapid variations in water characteristics, which allowed us to differentiate five zones. Water estimates with lower transparency and higher attenuation of radiation were found in the northern area of Alqueva reservoir during the months characterized by higher water temperatures, with Secchi depth monthly averages near 1.0 m and diffuse attenuation coefficient near or above 1.5 m$^{-1}$. Satellite retrievals of water with greater transparency in the reservoir were obtained in the southern area in months with low water temperature and atmospheric stability, presenting some monthly Secchi depth averages above 3 m, and diffuse attenuation coefficient below 0.8 m$^{-1}$. January 2018 presented great transparency of water with a Secchi depth of 7.5 m for pixels representing the 95th percentile and diffuse attenuation coefficient of 0.36 m for pixels representing the 5th percentile in the Southern region.

**Keywords:** 6SV; Alqueva reservoir; microalgae; water quality; Sen2Cor; turbidity

## 1. Introduction

Lake ecosystems are vital resources for aquatic wildlife and human needs, representing 98% of the liquid surface freshwater on the Earth's surface. Many organisms depend on freshwaters for survival, and humans frequently depend on lakes for many purposes such as drinking water, fisheries, agricultural irrigation, industrial services or recreation. In a warming climate associated with the increase of carbon dioxide concentration, changes are expected in precipitation patterns as the increase of extreme precipitation events, as well as the global increase in air temperature [1–5]. For southern

Portugal, Earth System models project a decrease in yearly mean precipitation (between -10% and -35 % for different scenarios) with an intensification of extreme precipitation, a significant increase of the maximum and minimum temperatures in all seasons and an increase in annual average number of heatwaves, much stronger and longer [6,7]. Lakes are sensitive to the impacts of climate change, and the impacts include a wide range of negative consequences, such as changes in thermal stratification, an acceleration of the eutrophication which favors periodic proliferation by cyanobacteria in many freshwaters, increased turbidity, and changes in salinity [8–10]. In this context, many studies have been carried out regarding the deterioration of water quality due to human influence, and its spatial and temporal variations in rivers or water reservoirs [11–14].

Alqueva reservoir is important for water supply and agricultural irrigation in the Alentejo (Southern Portugal), region with long periods of low precipitation associated with high values of air temperature, normally between May and September, and drought periods that can last more than 2 consecutive years [15]. Shifts in precipitation variability and seasonal runoff induce severe effects on water supply, water quality, and management of water resources in Southern Portugal. With warmer air temperatures expected for the Alentejo region, and consequently warmer surface waters in lakes, a favorable environment is created for the early growth of microalgae blooms, namely cyanobacteria, leading to an acceleration of the eutrophication [16,17].

The traditional field-based methods to monitor water quality in inland waters are usually costly and hardly allow adequate spatial and temporal analysis. Remote sensing could overcome these constraints, allowing to monitor water quality in large inland waters where conventional monitoring approaches tend to be limited [18]. From the launch of ERTS-1/Landsat-1 in August, 1972 up to and beyond the post-2000 launches of MODIS and MERIS, the mapping of water quality parameters using satellite images over inland lakes has exponentially increased [19–23]. Olmanson et al. [24] used the Landsat archive for mapping lake water clarity of over 10,000 Minnesota lakes, showing the great application to lakes of different dimensions and great agreement between satellite data and field measurements of Secchi depth (SD) within Landsat paths (range of $R^2$ between 0.71 and 0.96). In addition to the high spatial resolution, allowing the analysis of water quality in several reservoirs, remote satellite detection also allows for mapping water quality in lakes over the last 30 years [25]. The Sentinel-2 mission from the European Space Agency (ESA) is a land monitoring constellation of two satellites (S2A launched in June 2015 and S2B launched in March 2017) that provide high-resolution optical imagery (spatial resolution between 10 m and 60 m, depending on the spectral band) and presents a systematic global coverage of two to three days at mid-latitudes, which brought a great opportunity to study inland reservoirs. Several studies have been done in recent years in the analysis of water quality of inland waters using Sentinel-2 data, to measure water quality parameters of bodies of water [26–31].

Satellite data is used over lakes and reservoirs aiming to evaluate surface parameters, nevertheless, the satellite measures the signal of the surface plus the signal of atmosphere in-between. This unnecessary part of the signal is commonly removed using atmospheric correction codes that correct the top-of-atmosphere (TOA) signal, with respect to the atmospheric path, to obtain only the signal from the surface. In this study, the atmospheric correction is performed using two models: Sen2Cor and 6SV. Sen2Cor was developed by ESA for applications over land surfaces but it is also applied over water surfaces, with several studies presenting good estimates of surface reflectances used for monitoring water quality parameters in lakes and reservoirs, such as colored dissolved organic matter, chlorophyll *a*, turbidity, or SD [23,27,32]. The 6SV is an advanced radiative transfer code capable of accounting for radiation polarization in a mixed molecular–aerosol atmosphere being a vector version of the 6S [33]. The 6S atmospheric correction has already been successfully used in lakes and reservoirs [21,32,34–37].

Monitoring water quality using satellite remote sensing may involve relationships between surface reflectances in certain spectral bands (or band combinations) and water parameters analyzed in the laboratory or in situ. Potes et al. [21] developed a method to estimate concentrations of chlorophyll *a*

and cyanobacteria in the Alqueva reservoir, combining surface reflectances using the MERIS satellite and laboratory analyzes. In addition to these two parameters, the same author used MERIS satellite images to obtain estimates of turbidity in the same artificial lake [22]. In the past two years (2017 and 2018), microalgae blooms have been identified in the Alqueva reservoir through satellite images and in situ measurements, starting to develop in early summer in the northern region of Alqueva and gradually spreading through all reservoir. In October, these blooms tend to disappear with the normal decrease of air temperature and solar radiation, the beginning of precipitation and strong winds, associated with the arrival of the first Atlantic atmospheric fronts. Potes et al. [23] reported this situation, showing how an intense microalgae bloom vanished in early October of 2017 with the increase of water mixing due to the influence of hurricane Ophelia progressing along the Portuguese coast, causing significant precipitation, a decrease in the air temperature and inducing strong winds at Alqueva reservoir.

The main motivation of the present work is to explore reliable remote sensing methods for full spatial cover and continuous monitoring of key physical parameters, which affect the water quality of inland water bodies such as Alqueva reservoir. Within the multidisciplinary project ALOP (ALentejo Observation and Prediction systems), among other physical-chemical and biological variables, the Secchi depth and diffuse attenuation coefficient (KD) were measured, two parameters closely linked to the water transparency and the dissolved/suspended particles in the water. Nevertheless, this analysis is spatially and temporally limited. In order to derive reliable remote detection methods for continuous monitoring and adequate spatial coverage of physical parameters affecting the water quality of reservoirs such as Alqueva, data obtained from Sentinel-2 missions (Multi-Spectral Instrument (MSI)) was used. In addition to the high spatial resolution of 10-60 m, it presents a high re-visitation, with systematic global coverage in the Alqueva reservoir region of two to three days, considering the two twin satellites in operation. Landsat also has a high spatial resolution (30 m), but much shorter temporal resolution, with 16 days difference between each image, and this revisit time may be problematic for the detection of short events [38]. MODIS has high revisitation (every 1–2 days), but lower spatial resolution of 250–1000 m. Thus, Sentinel-2 represents the best choice for satellite monitoring of Alqueva reservoir compared with these two satellites.

SD and KD also control the vertical distribution of the absorption of solar radiation in the water column and thus play a relevant role in the air-water surface energy balance. For this reason, these parameters are used in numerical weather prediction, so their mapping from satellite remote sensing will contribute to improving the representation of lakes in operational forecasting systems, as shown in Potes et al. [22].

The main objective of this work is to develop a method based on satellite remote sensing to estimate SD and KD and subsequently analyze the spatial and temporal variations of these parameters in the period July 2017-June 2019 at Alqueva reservoir. These studies may also be relevant for the identification of areas with different patterns of water characteristics and their categorical relation with meteorological data.

## 2. Study Area and Data

### 2.1. Study Area

The study area presented in this paper is the Alqueva reservoir, located in the south of Portugal (Alentejo region) along 83 km of the main course of the Guadiana River. It constitutes the largest artificial lake in the Iberian Peninsula with a capacity of 4.150 hm$^3$ and a surface area of 250 km$^2$ at the full storage level of 152 m.

According to Köppen climate classification, this region is identified by type Csa—hot summer Mediterranean climate, featured by a temperate climate with dry and hot summers. The Alentejo region is characterized by relevant irregularity of inter-annual precipitation, with average accumulated precipitation of the order of 500 mm. It is normal that this region presents two summer months, July and

August, with nearly no precipitation and maximum temperatures above 30 °C in most of the days, and heat waves with temperatures near or greater than 40 °C, accompanied by very low relative humidity. The ALOP project aims to develop multifunctional activities in the field of interactions between the atmosphere, water and ecosystems, covering, among other tasks, meteorological observation. Three meteorological stations were installed in 2017, MontanteP station on a platform in the middle of the reservoir, and two stations on both margins, BarbosaM and Cid AlmeidaM (Figure 1). Figure 2 shows the great irregularity in accumulated monthly rainfall presenting periods of several consecutive months with very scarce precipitation (less than 25–50 mm). Data is obtained from a meteorological station, installed in March 2017 (Cid AlmeidaM station), in the East margin of the reservoir.

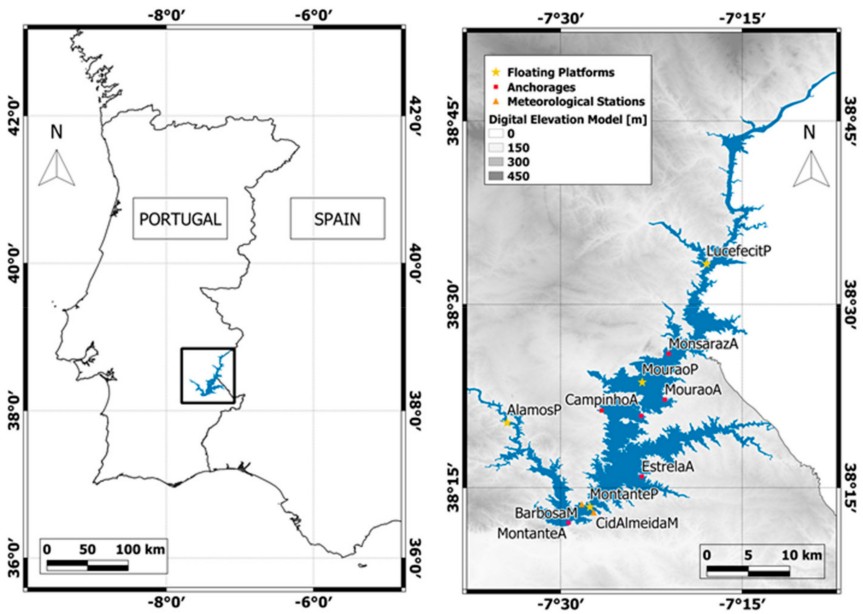

**Figure 1.** Alqueva reservoir and position of the sites used in this study.

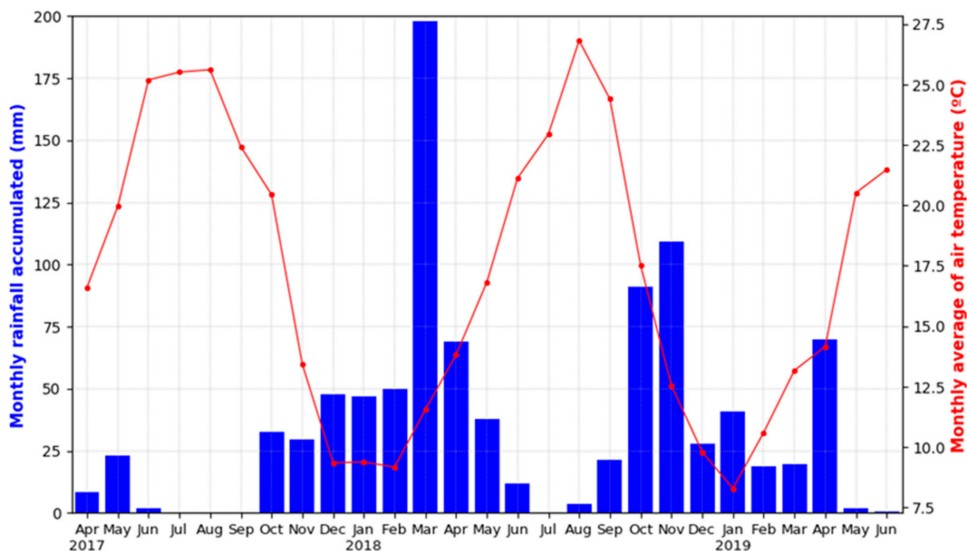

**Figure 2.** Monthly rainfall and air temperature in Alqueva for the period of April 2017–June 2019. Data from Cid Almeida meteorological station.

*2.2. Data Collection*

2.2.1. Diffuse Attenuation Coefficient and Secchi Depth Measurements

Since February 2016 periodic measurements of KD and SD have been made in the Alqueva reservoir. The measurements were carried out approximately every 1-2 months at various points in the Alqueva reservoir to monitor their temporal and spatial variations. Figure 1 shows the map with the location of the measurement sites. AlamosP, MontanteP, MouraoP and LucefecitP represent floating platforms installed in the deeper areas of the reservoir, and far from the margins. MontanteA, EstrelaA, LuzA, CampinhoA and MonsarazA represent anchorages, located in the margins of the reservoir.

In order to ensure adequate illumination, these measurements were limited to the daylight period between 10:00 a.m. and 4:00 p.m. UTC. In this study, the measurements of KD in water were made using the method described in Potes et al. [39]. According to this method, the measurements of underwater downwelling spectral irradiance were performed with the use of a portable FieldSpec (Analytical Spectral Devices; Boulder, CO, USA) together with an optical fiber linked to a hemispherical tip (180°). Then, an exponential fit was applied through the spectral profile and depth to obtain the spectral diffuse attenuation coefficient (KD). The spectral KD, with 1 nm resolution, is averaged for the spectral region 400–700 nm in order to obtain a value corresponding to the Photosynthetically Active Radiation (PAR) region, thus the KD used in this work is the KD (PAR). SD was measured only on clear days, at the shaded side of the anchorages or platforms to avoid direct sunlight reflections from the water surface. Two SD values were measured, each by a different person, to minimize deviations due to the observer, and the averaged SD value was used. A slight deviation between each SD measurement of approximately 0.14 m on average was observed.

The SD measures the transparency of the water and the values obtained in the study period ranged between 0.4 m and 7 m. KD measures the absorption and dispersion of the radiation by the particles present in the water, and the values ranged between 0.4 $m^{-1}$ and 2.4 $m^{-1}$. The extreme values of low SD and high KD in the water coincide with a microalgae bloom in September 2017. Otherwise, the values of SD and KD related with the highest transparency were recorded in periods without rainfall, with a stable atmosphere and water temperature below 22 °C. The value of 7.0 m for SD was measured on 1 March 2019 and this measurement was by far the highest SD, with a previous maximum of 5.0 m. This impressive record for this reservoir should have been caused by a set of circumstances, such as the little precipitation in the previous month (February presented precipitation below average and no days of intense precipitation), mild temperatures (maximum between 15 °C and 20 °C), and very low wind speed both in the measurement day and in the previous days.

2.2.2. Sentinel-2 data

The Sentinel-2 mission consists of a constellation of two polar-orbiting satellites, Sentinel-2A and Sentinel-2B, each one equipped with an optical imaging sensor (MSI). Sentinel-2A was launched on 23 June 2015 and Sentinel-2B followed on 7 March 2017. These twin polar-orbiting satellites allow a high 2-3 days revisit time for Alqueva reservoir since July 2017. MSI data are acquired in 13 spectral bands in the visible and near-infrared and have very high spatial resolution, with three bands at 60 m, six bands at 20 m and four bands at 10 m (Table 1). The dataset used in this study is the Sentinel-2 Level-1C product, which is composed of 100 km × 100 km tiles in the UTM/WGS84 projection and provides the Top-Of-Atmosphere (TOA) reflectance. The Sentinel-2 tiles were downloaded from the Copernicus Open Access Hub, free and open access to Sentinel-2 data in SENTINEL-SAFE format (https://scihub.copernicus.eu/dhus/#/home).

**Table 1.** The central wavelength, bandwidth and spatial resolution of the 13 bands of the Multi-Spectral Instrument (MSI) instruments onboard Sentinel-2A and 2B.

| Band Number | S2A | | S2B | | |
| --- | --- | --- | --- | --- | --- |
| | Central Wavelength (mm) | Bandwidth (mm) | Central Wavelength (mm) | Bandwidth (mm) | Spatial Resolution (m) |
| 1 | 442.7 | 27 | 442.2 | 45 | 60 |
| 2 | 492.4 | 98 | 492.1 | 98 | 10 |
| 3 | 559.8 | 45 | 559.0 | 46 | 10 |
| 4 | 664.6 | 38 | 664.9 | 39 | 10 |
| 5 | 704.1 | 19 | 703.8 | 20 | 20 |
| 6 | 740.5 | 18 | 739.1 | 18 | 20 |
| 7 | 782.8 | 28 | 779.7 | 28 | 20 |
| 8 | 832.8 | 145 | 832.9 | 133 | 10 |
| 8a | 864.7 | 33 | 864.0 | 32 | 20 |
| 9 | 945.1 | 26 | 943.2 | 27 | 60 |
| 10 | 1373.5 | 75 | 1376.9 | 76 | 60 |
| 11 | 1613.7 | 143 | 1610.4 | 141 | 20 |
| 12 | 2202.4 | 242 | 2185.7 | 238 | 20 |

## 3. Methodology

The process of developing an empirical remote sensing model for monitoring of spatial and temporal variations of SD and spectral KD during the period July 2017 - June 2019 was preceded by application of an atmospheric correction to images of the top of the atmosphere (TOA) in order to obtain surface spectral reflectance. Then the atmospherically corrected spectral reflectances obtained were validated (Section 3.1) and subsequently, regression techniques were used to develop relationships between field-measured parameters and water reflectance (Section 3.2).

### 3.1. Atmospheric Correction Validation

Satellite remote sensing of the surface in the visible and near-infrared is strongly affected by the presence of the atmosphere. Due to the low reflectance of water surfaces, the accurate removal of atmospheric effects is essential. Initial satellite data is provided at the top of the atmosphere (TOA), which is affected by atmospheric constituents through absorption and scattering processes. However, the analysis of water surface properties from remote sensing techniques requires the removal of the atmospheric effects, to obtain surface reflectances (or radiances). In the visible spectral region, in the absence of clouds, the main atmospheric effects that need to be corrected are aerosols, water vapor and ozone. In the solar spectrum, the atmospheric gaseous absorption is principally due to oxygen, carbon dioxide, methane, nitrous oxide, ozone and water vapor, but only the last two are not constant and depend on the time and location. The water vapor contribution affects mainly wavelengths greater than 700 nm and ozone presents a significant absorption between 550 and 650 nm. Therefore, the effect of ozone and water vapor can be accounted for since they have a great influence on the visible region, between 400 nm and 750 nm, the region of the spectrum used to estimate surface parameters at Alqueva reservoir.

In this paper, two atmospheric correction methods were assessed, applied to images acquired by the Multi-Spectral Instrument (MSI) on-board the European Space Agency's Sentinel-2A and Sentinel-2B:

i) 6SV which is a vector version of the 6S (Second Simulation of a Satellite Signal in the Solar Spectrum) radiative transfer code [40].

ii) Sen2cor, which is a processor provided by ESA for Sentinel-2 Level 2A product generation [41].

6SV is a new version of old 6S and is available at http://6s.ltdri.org and a summary of the code validation can be found in Kotchenova et al. [42] and in Kotchenova et al. [43]. The 6SV is an advanced radiative transfer code developed by the MODIS LSR SCF (Land Surface Reflectance Science Computing Facility).

The atmospheric correction processor Sen2Cor was developed by Telespazio VEGA Deutschland GmbH on behalf of ESA. Sen2Cor is a Level-2A processor to correct the Sentinel-2 Level-1C Top-Of-Atmosphere (TOA) products for the effects of the atmosphere in order to deliver a Level-2A Bottom-Of-Atmosphere (BOA) reflectance product. ESA not only provides the bottom of atmosphere reflectance based on the implementation of the Sen2Cor algorithm, but also additional products as Quality Indicators for cloud and snow probabilities, Aerosol Optical Thickness (AOT), Water Vapour (WV) and Scene Classification (SC). Obregón et al. [44] presented a validation of aerosol optical thickness (AOT) and integrated water vapor (IWV) products provided by ESA obtained from Sentinel MSI, and for this purpose used a significant number of stations over Europe and adjacent regions, between March 2017 and December 2018. The results showed the high reliability of the WV estimates, with normalized root mean square errors (NRMSE) of 5.33% and $R^2$ of 0.99. The comparison showed lower agreement, nonetheless reliable, for AOT with NRMSE of 9.04% and $R^2$ of 0.65. The AOT at 550 nm and WV provided by ESA Sen2Cor were used here as inputs for the 6SV radiative transfer code.

The TOA reflectances were all resampled to 60 m resolution, with downsampling of 10 m and 20 m data (Table 1) by block averaging and corrected using the 6SV considering the corresponding spectral response functions characterizing the Sentinel-MSI bands. A new version of the Sentinel-2A spectral functions, correcting the responses for bands 1 and 2 released after 15 January 2018 (https://earth.esa.int/web/sentinel/missions/sentinel-2-news/; last accessed 21/07/2019) was used. 6SV accounts for the adjacency effects due to reflection from contiguous pixels. However, in almost all its area, the reservoir is wide comparatively to MSI spatial resolution, and surrounded by soil or very low vegetation. We have assumed that the surface (all water pixels in the reservoir) behave as a homogeneous Lambertian reflector. 6SV accounts for a wide variety of sensor characteristics and atmospheric conditions. The 6SV used to correct the MSI images over Alqueva region, allows to account for the various effects of the atmosphere, namely the ozone and water vapor columns, aerosol optical thickness (AOT) at the reference wavelength of 550 nm, aerosol characterization (type and concentration), spectral/geometrical conditions, and in addition ground reflectance (type and spectral variation). The input parameters for the 6SV process to model the atmospheric effects are shown in Table 2.

**Table 2.** Input parameters of 6SV model applied to Sentinel-2 MSI.

| | Source | Parameters |
|---|---|---|
| Input type | Sentinel-2 MSI | TOA reflectance |
| Geometrical Conditions | Sentinel-2 MSI | Solar Zenith angle, Solar Azimuth angle (°) |
| | | View Zenith angle, View Azimuth angle (°) |
| | | Month, Day |
| Atmospheric Conditions (User) | Product of SEN2COR | Water vapour (g/cm$^2$) |
| | Ozone Monitoring (OMI) | Ozone (cm-atm) |
| Aerosol Model (Type) | - | Continental |
| Aerosol Model (Concentration) | Product of SEN2COR | Aerosol Optical Thickness at 550 nm |
| Effect of the altitude | Water level | Target (km) |
| | - | Sensor aboard a Satellite |
| Spectral bands | Sentinel-2 MSI | Spectral function responses |
| Ground Reflectance | - | Homogeneous Lake Water |

Satellite-derived water spectral reflectances were compared with surface spectral reflectance measurements at Alqueva selected sites, obtained with a portable spectroradiometer FieldSpec UV/VNIR (ASD, Inc), aiming at validating the results obtained with the atmospheric correction methods. In situ measurements were performed as close as possible to the satellite overpass (maximum difference of 1 day).

The spectroradiometer measures the intensity of the light field across a given point, so reflectance is calculated as the ratio between the energy leaving the sample by reflection and the energy incident on the sample (obtained from the measurement with the white reference panel). Since the calculation is a ratio, energy units cancel out. In fact, to calculate reflectance with the spectroradiometer, although it is calibrated, there is no need for radiometric calibration. The electrical current signals are converted into computer type digital numbers (DN). The reflectance is then calculated as the ratio between the raw DN of the energy reflected from the sample and the raw DN of the energy incident on the sample (obtained from white reference measurement) as described by ASD [45]. Measurements were made with the 10° field-of-view. In order to ensure correct reflectance measurement, the illumination should be practically the same for the lake and white reference measurements, so the measurement of the white plate was made immediately prior to the measurement of the lake, in order to avoid large variations of sun incidence in both measurements. In situ values were obtained by averaging 25 spectra selected to represent the Alqueva water reflectance. The mean satellite reflectance was calculated using a $2 \times 2$ pixel box close to each sampling station to perform a direct comparison with in-situ measurements. Statistical indicators have been used in this study to obtain an appropriate assessment of the atmospheric correction methods: The Bias error, which is a measure of the overall error or systematic error, check the average bias of the data, e.g., if positive indicates that the estimated values are overestimated compared to those observed and if negative are underestimated:

$$\text{Bias} = \frac{1}{N} \sum (X - X_{meas}),\tag{1}$$

where N corresponds to the total number of data, X are estimations of satellite data and $X_{meas}$ corresponds to in situ measurements.

The mean absolute error (MAE) is an average of the absolute errors:

$$\text{MAE} = \frac{1}{N} \sum \left| (X - X_{meas}) \right|.\tag{2}$$

The mean absolute percentage error (MAPE) is defined as the mean of absolute differences between estimates (satellite-derived) and observations and measures the size of the error in percentage terms:

$$\text{MAPE} = \frac{1}{N} \sum \left| \frac{X_{meas} - X}{X_{meas}} \right| \times 100.\tag{3}$$

The root mean square error (RMSE) gives a measure of the accuracy existing between estimated values (satellite retrievals) and the corresponding surface measurements:

$$\text{RMSE} = \sqrt{\frac{1}{N} \sum (X_{meas} - X)^2}.\tag{4}$$

The normalized value of the previous statistic indicator RMSE has been calculated to compare estimations of satellite and ground-based measurements, where a small value of NRMSE identifies a good agreement with the field observations:

$$\text{NRMSE } (\%) = \frac{RMSE}{mean(meas)} \times 100,\tag{5}$$

where mean (meas) corresponds to the average of observations.

The Nash–Sutcliffe efficiency (NSE) is a normalized statistic that determines the relative magnitude of the residual variance compared to the measured data variance:

$$\text{NSE} = 1 - \frac{\sum (X_{meas} - X)^2}{\sum \left( X_{meas} - mean(meas) \right)^2}.\tag{6}$$

Another statistical indicator used is the correlation coefficient which indicates the direction and strength of a linear relationship between two variables. This parameter, which shows how much the model can describe the data, should be close to 1 for an ideal situation of perfect match between both.

Figure 3 shows a comparison between the measured and satellite-derived water reflectance for the first five bands, with 6SV and Sen2Cor. Tables 3 and 4 shows the statistical indicators obtained for each of the two atmospheric correction methods considered.

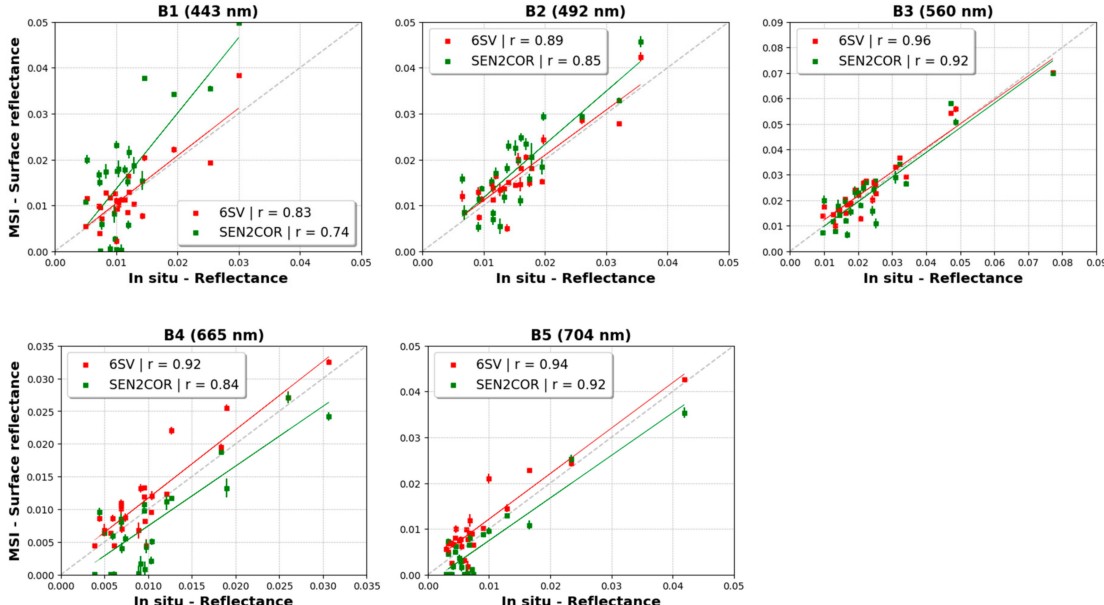

**Figure 3.** Comparison between in situ and MSI reflectances after atmospheric correction. The dashed line indicates the y = x, the solid lines (red and green) show the best-fits and the vertical error bars represent the standard deviations corresponding to the four selected pixels.

**Table 3.** Statistical indicators between the measured and satellite-derived water reflectance for the 6SV method.

| N= 25 | Bias | MAPE (%) | Correlation | RMSE | $R^2$ | NSE |
|-------|------|----------|-------------|------|-------|-----|
| B1 | 0.0005 | 29 | 0.83 | 0.0039 | 0.69 | 0.51 |
| B2 | 0.0010 | 22 | 0.89 | 0.0036 | 0.79 | 0.72 |
| B3 | 0.0014 | 19 | 0.96 | 0.0044 | 0.92 | 0.91 |
| B4 | 0.0017 | 30 | 0.92 | 0.0034 | 0.84 | 0.72 |
| B5 | 0.0022 | 51 | 0.94 | 0.0037 | 0.88 | 0.79 |

**Table 4.** Same as in Table 3 but for the Sen2Cor method.

| N= 25 | Bias | MAPE (%) | Correlation | RMSE | $R^2$ | NSE |
|-------|------|----------|-------------|------|-------|-----|
| B1 | 0.0046 | 85 | 0.74 | 0.0102 | 0.55 | -2.29 |
| B2 | 0.0026 | 36 | 0.85 | 0.0056 | 0.73 | 0.33 |
| B3 | -0.0007 | 23 | 0.92 | 0.0059 | 0.85 | 0.83 |
| B4 | -0.0026 | 44 | 0.84 | 0.0047 | 0.70 | 0.47 |
| B5 | -0.0024 | 54 | 0.92 | 0.0040 | 0.85 | 0.75 |

Sen2cor method presents several cases of null reflectances for band 1 (443 nm), band 4 (665 nm) and band 5 (704 nm), and generally worse statistical indicators than the radiative transfer code 6SV. The cases of null reflectances are often associated with situations of very clean waters causing the Sen2Cor to excessively correct the visible bands (Figure 3). Martins et al. [32] also reported this situation with MSI-corrected reflectances using Sen2Cor, obtaining a better match with in situ measurements over brighter lakes, rather than dark lakes.

With 6SV method, the bias is positive for all bands and the red solid line always above the dashed line y = x, denoting on average an overestimation of the calculated reflectances in relation to the measurements. In contrast, with Sen2Cor atmospheric correction process, there is overestimation in bands 1 and 2, and underestimation in the remaining bands of the visible region. Note the large error associated with the Sen2Cor process in band 1 of the MSI instrument, which despite having some null reflectances, has a positive bias, with several estimated values much higher than those observed, with a very high MAPE of 85% and worst correlation, RMSE and bias for this region of the spectrum. The very weak accuracy of the Sen2Cor method in band 1 is also proven by NSE = - 2.3, which, being lower than 0, indicates unreliable satellite estimations. Regarding the Sen2Cor atmospheric correction process only in band 3 (NSE = 0.83) and band 5 (NSE = 0.75) the satellite data fits with some accuracy the observations. As for the 6SV atmospheric correction, excluding band 1 from MSI (NSE = 0.51), this method features good accuracy, with NSE greater than 0.7 for all analyzed bands, indicating a good match between estimates and observed reflectances. The atmospheric correction performed using the 6SV method presents better statistical results compared to Sen2Cor for all analyzed bands, e.g., higher correlation and lower MAPE values, smaller deviations from the straight line to line y = x (also showing a lower bias) and even much lower RMSE values. With 6SV there are higher correlations, greater than 0.8 for all bands, and values lower than or equal to 30% of MAPE, except MSI band 5, with an associated MAPE of 51%. Excellent results are obtained for band 3 with a correlation coefficient of 0.96, MAPE below 20%, and NSE of 0.91, the nearest value to the perfect match of estimated vs observed data (NSE = 1). This excellent correction of the 6SV method to band 3 is essential, because high microalgae densities are present in the Alqueva reservoir for several months, which yield high reflectances in this region of the spectrum.

It is verified that the best atmospheric correction method for the Alqueva reservoir is achieved with the 6SV method using the WV and AOT 550 products from Sen2Cor. This method was then used to accurately estimate water parameters.

## 3.2. Empirical Algorithms

In situ measurements of SD and KD were related to the atmospherically corrected surface spectral reflectances obtained from MSI (using 6SV - Section 3.1), through regression algorithms. The average reflectance of the four closest pixels to the measurement site was calculated. The time gap between in-situ measurements and the satellite overpass affects the reflectance comparison and, in this context, several studies report between 3 to 8 days as the maximum time lag used between on-site measurements and satellite images for a correct comparison, with stable meteorological conditions and in the absence of algal blooms [24,27,46]. In the present study, due to the presence of algae in a large period of the year, and in order to reduce the error associated with rapid variations in water characteristics, only satellite images with maximum differences of 1 day were selected in relation to in situ measurements [47]. This time difference is considered a reasonable compromise for a reservoir of large dimensions as Alqueva, aiming at obtaining a dataset that ensures the robustness of the empirical algorithms. The data used to develop the regression algorithms refers to the period of April 2016–October 2017. The data used for the validation corresponds to a different period, between November 2017 and June 2019. The best algorithm of SD was found empirically by deriving a regression model for different combinations of bands in the visible region, selecting the best fit according to the first, with the greatest $R^2$ and second, with the lowest values of NRMSE. Only bands in the visible region were used, up to 704 nm (corresponding to band 5 of the MSI instrument), because the Alqueva reservoir has several days with very low reflectance for higher wavelengths, corresponding mainly to periods of the year and areas with low turbidity and biological activity. Several empirical algorithms available from the literature were tested for the Alqueva reservoir, with a poorer performance with respect to the algorithm proposed here as shown in Table 5. Rotta et al. [48] proposed an algorithm for KD retrieval applied to Nova Avanhandava Reservoir, using the 660 nm band. In this work the equivalent MSI band (band 4) is used, adjusting the linear regression coefficients to Alqueva reservoir, yielding Equation

(8). The results of $R^2$ and NRMSE obtained for the Alqueva reservoir with the algorithm proposed by Rotta et al. [48] are presented in Table 6.

**Table 5.** Statistical parameters obtained for the application of Secchi Depth algorithms from the literature and that proposed in this study, to the Alqueva reservoir.

| Algorithms | Equation | $R^2$ | NRMSE (%) |
|---|---|---|---|
| Verdin, J.P. [49] | $\frac{1}{(0.665+35.6\times B4)}$ | 0.69 | 28 |
| Lavery et al. [50] | $2.5 - (0.56 \times B4) - (0.42 \times \frac{B2}{B4})$ | 0.60 | 53 |
| Wu et al. [51] | $EXP(1.3-(0.27\times B2)-(0.65\times B4))$ | 0.69 | 71 |
| Bonansea et al. [52] | $1.25 - 0.44\times B4 + 0.11\times \frac{B2}{B4}$ | 0.65 | 84 |
| Rotta et al. [48] | $2.0709 \times\left(\frac{B3}{B4}\right)$ -1.2697 | 0.1 | 72 |
| Jesús Delegido et al. [53] | $4.7134 \times (\frac{B2}{B3})^{2.5569}$ | 0.29 | 51 |
| Page et al. [54] | $EXP(2.437\times\frac{B2}{B4} - 2717.82 \times$ $(B4 \times B5) - 2.469)$ | 0.56 | 102 |
| **Proposed algorithm** | $0.024 \times \frac{B2}{B3\times B4} + 0.72$ | 0.86 | 17 |

**Table 6.** Same as in Table 5 but for the diffuse attenuation coefficient.

| Algorithms | Equation | $R^2$ | NRMSE (%) |
|---|---|---|---|
| Rotta et al. [48] | $67.4155 \times B4 +0.3978$ | 0.81 | 33 |
| **Proposed algorithm** | $54.3 \times B4 + 0.32$ | 0.82 | 21 |

The scatter plots, with the best fit for each algorithm, are shown in Figure 4. Both relations are of linear type, with high coefficients of determination: $R^2$ of 0.86 for the SD algorithm and $R^2$ of 0.82 for the KD algorithm. The NRMSE is equal to 17% and 21% for the SD and KD algorithms respectively.

$$SD = \left(0.024 \times \frac{B2}{B3 \times B4}\right) + 0.72 \tag{7}$$

$$KD = 54.3 \times B4 + 0.32. \tag{8}$$

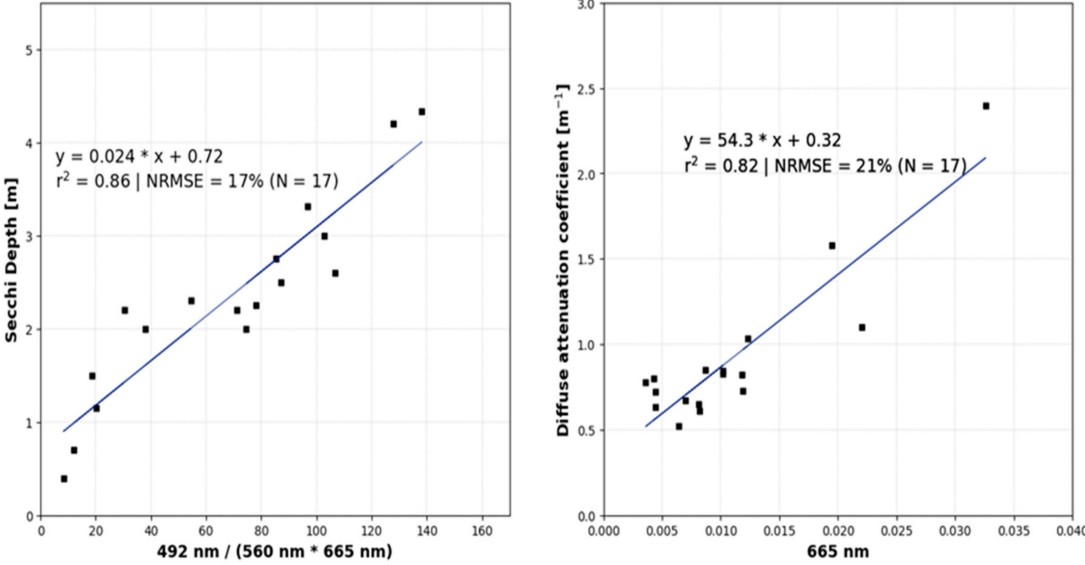

**Figure 4.** Scatter plot between parameters of water quality (SD on the left and KD on the right) and the bands of MSI. Shown also is the linear regression obtained.

## 4. Results and Discussion

*4.1. Validation of Algorithms of Water Quality Parameters*

The results obtained with the algorithms presented in Section 3.2 were validated for a different period, between November 2017 and June 2019, through the comparison of the satellite retrievals with the measurements obtained from in situ water sampling.

The algorithms presented in Section 3.2 were applied to the four nearest pixels with respect to the sites where the measurements were taken and the mean value was computed. In Figure 5 the relationships between estimated SD/KD and the corresponding data measurements, is shown.

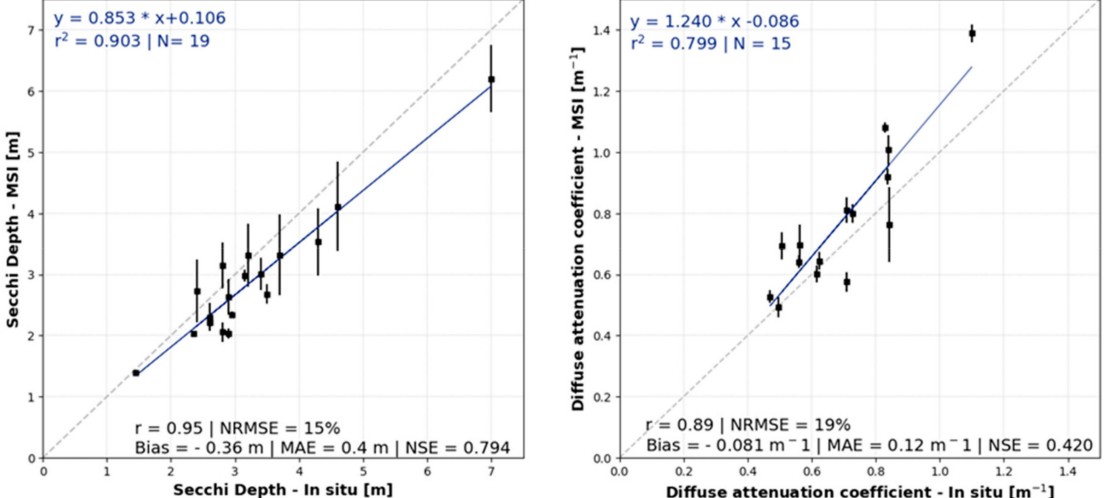

**Figure 5.** Scatter plot of predicted versus observed SD (left) and KD (right). The dashed line indicates the y=x, and solid line (blue) show the best-fit lines. The vertical error bars represent the standard deviations corresponding to the four selected pixels.

A good agreement is verified between estimations and measurements, presenting high correlation coefficient values, with r = 0.95 and r = 0.89 for SD and KD respectively. Data spread is also small, with NRMSE of 15% for SD algorithm and NRMSE of 19% for KD algorithm. The high NSE for the SD algorithm (NSE = 0.794) indicates a good fit between observations and estimations. The MAE shows good results, with 0.4 m for SD and 0.12 $m^{-1}$ for KD. This means that on average there was a deviation of less than half a meter for SD, which is a small deviation value if considered that the validation values range between 1.5 m and 7.0 m. As for the KD, there is also a reduced MAE for the range of measured values. Most of the errors associated with the measurements are overestimated for the KD (Positive bias) and underestimated for SD, of 0.081 $m^{-1}$ and - 0.36 m respectively. In the validation of the SD algorithm, it is verified that the absolute value of the Bias (0.36 m) is practically equal to the MAE (0.4 m), meaning that deviations to the observed data are practically all of underestimation satellite retrievals in relation to the observed data. The validation of the KD algorithm only presents 2 points below the y=x line, which demonstrates that for almost all cases analyzed, when there is a deviation from satellite estimates in relation to the observations, data are overestimated. Thus we can conclude that although the average associated errors are reduced, a lower transparency of water (Lower SD) is expected from the atmospheric correction method presented for Alqueva reservoir and a lower attenuation of radiation in water (Higher KD) compared to observation data.

*4.2. Relation between Secchi Depth/Diffuse Attenuation Coefficient and Microalgae Bloom*

In addition to validating the algorithms (comparing with independent data) it is also important to investigate how the SD and KD vary spatially in the entire reservoir. One of the useful and relevant applications of Sentinel-2 is the identification and monitoring of rapid variations of microalgae

blooms in the lakes through, for example, RGB image composites using three bands of MSI with 10 m of resolution: 492 nm (blue), 560 nm (green) and 665 nm (red). Figure 6 presents two examples of RGB images showing the beginning of a microalgae bloom development in July 2018 and the presence of microalgae in most of the reservoir in the end of September 2018 with measurements of chlorophyll *a* and turbidity of 10.19 μg/L and 2.22 NTU, respectively, on 6 September 2018 (nearest in situ measurement available). Chlorophyll *a* determination was based in molecular absorption spectroscopy and the equations developed by Lorenzen [55], according to the standard methods NP 4327:1996 [56] and EN ISO 10260:1992 [57]. Turbidity was determined using the nephelometric method [58].

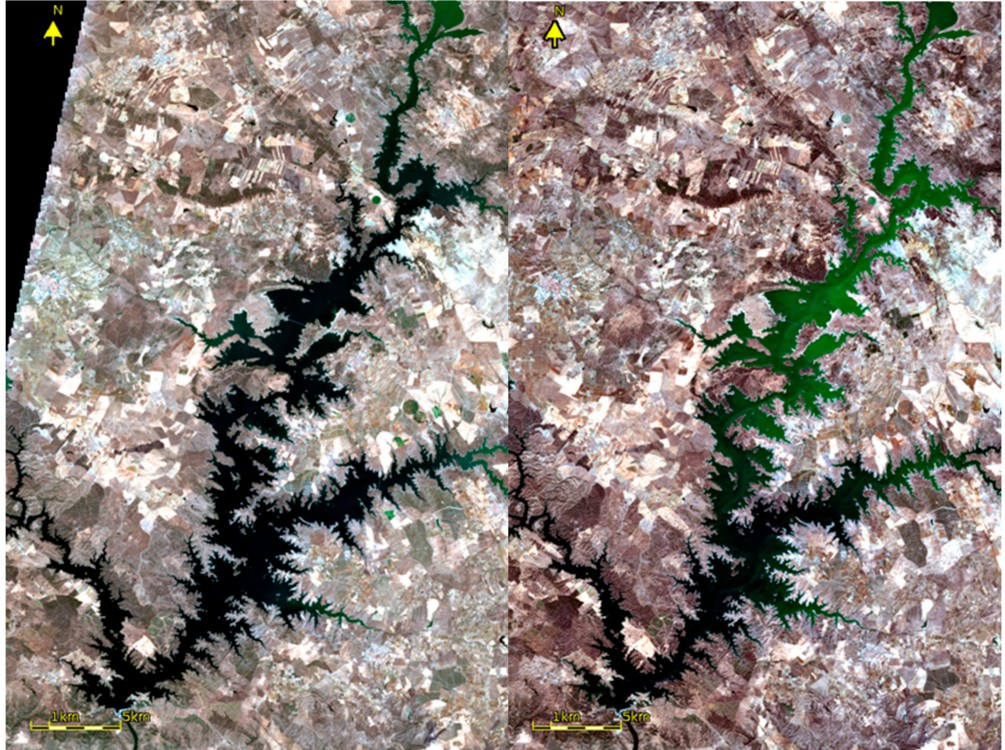

**Figure 6.** RGB Image showing the evolution of microalgae bloom for 16 July 2017 (left) and for 22 September 2017 (right) in the Alqueva reservoir.

Figure 7 represents the spatial distribution of SD and KD for 22 September 2017 obtained using the algorithms developed. The presence of algae in water tends to decrease the transparency of water. Therefore, under these conditions, a decrease in SD (associated with decreased water transparency) and an increase in the KD of water is expected due to an increase of solar radiation extinction in the water column (absorption plus scattering). This is precisely what is shown in Figure 7, with high KD and low SD values in the northernmost area of the reservoir, where the presence of algae in the RGB image (Figure 6) is evident, whilst there is no evidence of microalgae in the RGB image in the south and west of the reservoir. Figure 7 shows very high values of KD and low SD between Campinho and Monsaraz (see Figure 1), and on the right margins of the reservoir. The areas in the westernmost branch and south area of the reservoir, presented the highest SD and lowest KD values, with values of SDs greater than 2 m and KDs less than 1.0 m$^{-1}$. This will probably be associated with the non-propagation of the algae bloom to these locations, thus having smaller dissolved and suspended particles, more transparent water (greater SD) and less extinction of solar radiation in the water column.

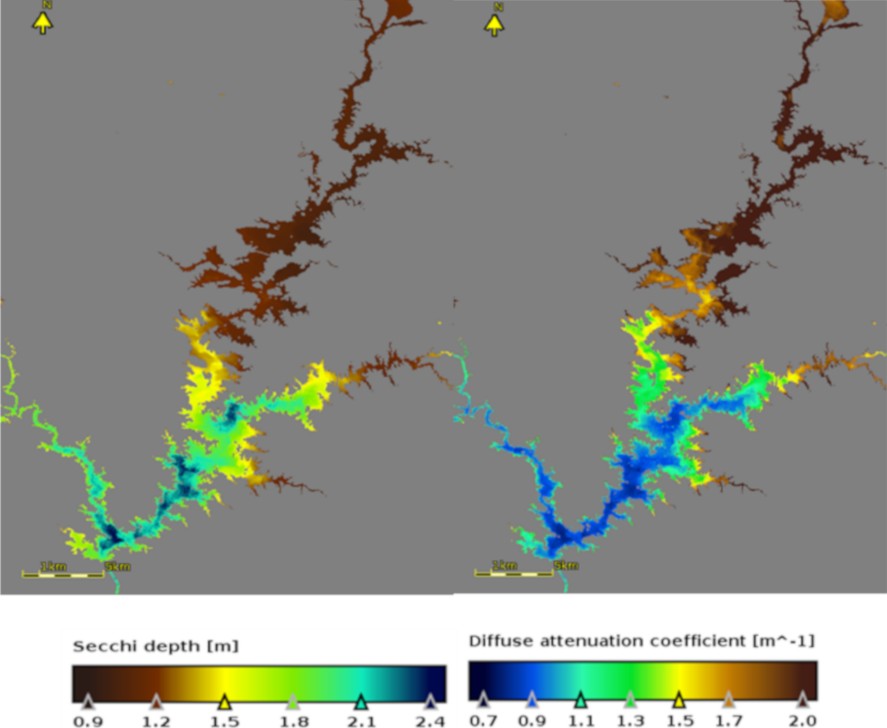

**Figure 7.** Secchi depth (left) and diffuse attenuation coefficient (right) maps for the Alqueva reservoir on 22 September 2017.

### 4.3. Seasonal and Spatial Distribution

The seasonal variation of the studied water parameters in Alqueva reservoir was evaluated using all Sentinel-2 MSI data available between July 2017 and June 2019. During this period and considering the use of the two satellite (S2A and S2B) overpasses in Alqueva region, it was possible to achieve at least 1 usable image per month. After atmospheric correction of the satellite images following the methodology described in Section 3.1, the empirical relations obtained (Equations (7) and (8)), were applied to the atmospherically corrected satellite measurements, obtaining full coverage of daily SD and KD in the reservoir during the period July 2017-June 2019.

In order to obtain a good distribution of the variables under study, an accurate identification of water pixels is essential. One of the advantages of the Sen2Cor output is the Scene classification (SC) product used to classify pixels, e.g., water, land, clouds. In this work, only water pixels were extracted from the images and used for atmospheric correction with 6SV. Occasional errors were found in the Sen2Cor classification, with water classification attributed to some pixels presenting effects of sunglint or cloudy shades. Various water body mapping approaches have been developed to identify water bodies in multispectral images. The Modified Normalized Difference Water Index (MNDWI) uses the green and Shortwave-Infrared (SWIR) bands and is one of the most popular methods [59]. In summary, to have a better mask of water pixels in the Alqueva reservoir, two conditions were imposed: Water scene Classification from Sen2Cor and positive values of MNDWI.

As for the seasonal analysis, the year was divided into 3 periods: July-October (JASO), November-February (NDJF), and March–June (MAMJ). JASO corresponds to the period of the highest air and water temperatures and with a great probability of microalgae blooms occurrence; NDJF is the coldest period and MAMJ is the period where the reservoir starts stratification.

For spatial characterization, the reservoir was divided into different areas with similar patterns of water characteristics. Only days with 100% of water pixels in the reservoir were considered, in order to calculate SD and KD averages counting the same sample of days for each pixel of the reservoir. In order to find patterns of water characteristics, not only the mean values are important, but also the

values corresponding to the days with extremes of SD and KD. Thus, the values corresponding to the 5th percentile and 95th percentile of SD and KD for each pixel were calculated.

It should be noted that the 5th percentile represents days with lower water transparency for SD and less attenuation of solar radiation in water for KD (for lower KD, more solar radiation penetrates in the water column, meaning more clear water). Figure 8 presents the 5th percentile (left panels), average (middle panels) and 95th percentile (right panels) of SD for the three different periods and Figure 9 presents the same parameters for the KD.

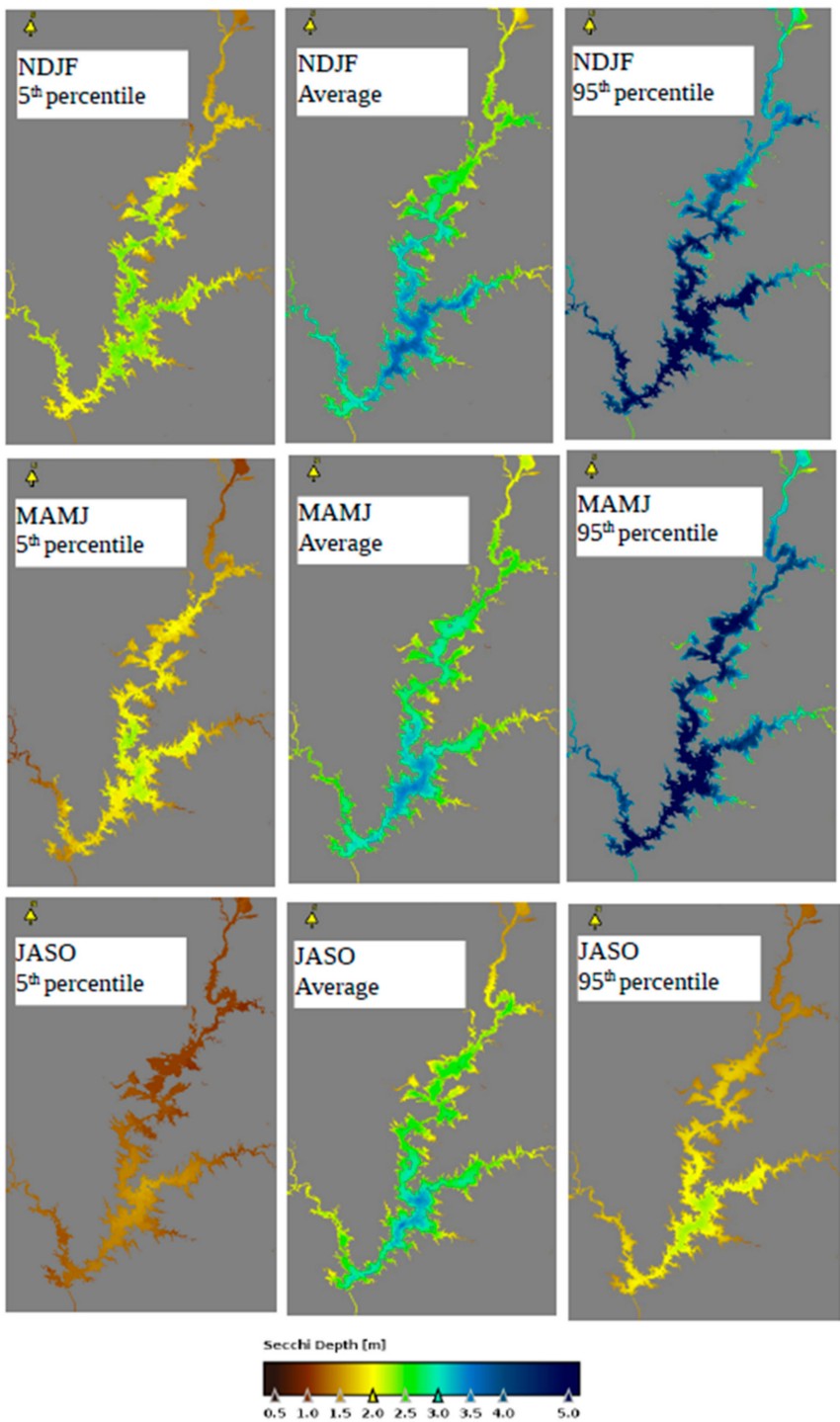

**Figure 8.** Spatial distribution of Secchi Depth shown from November to February (top), March to June (middle), and July to October (bottom) in Alqueva reservoir, for the period July 2017–June 2019.

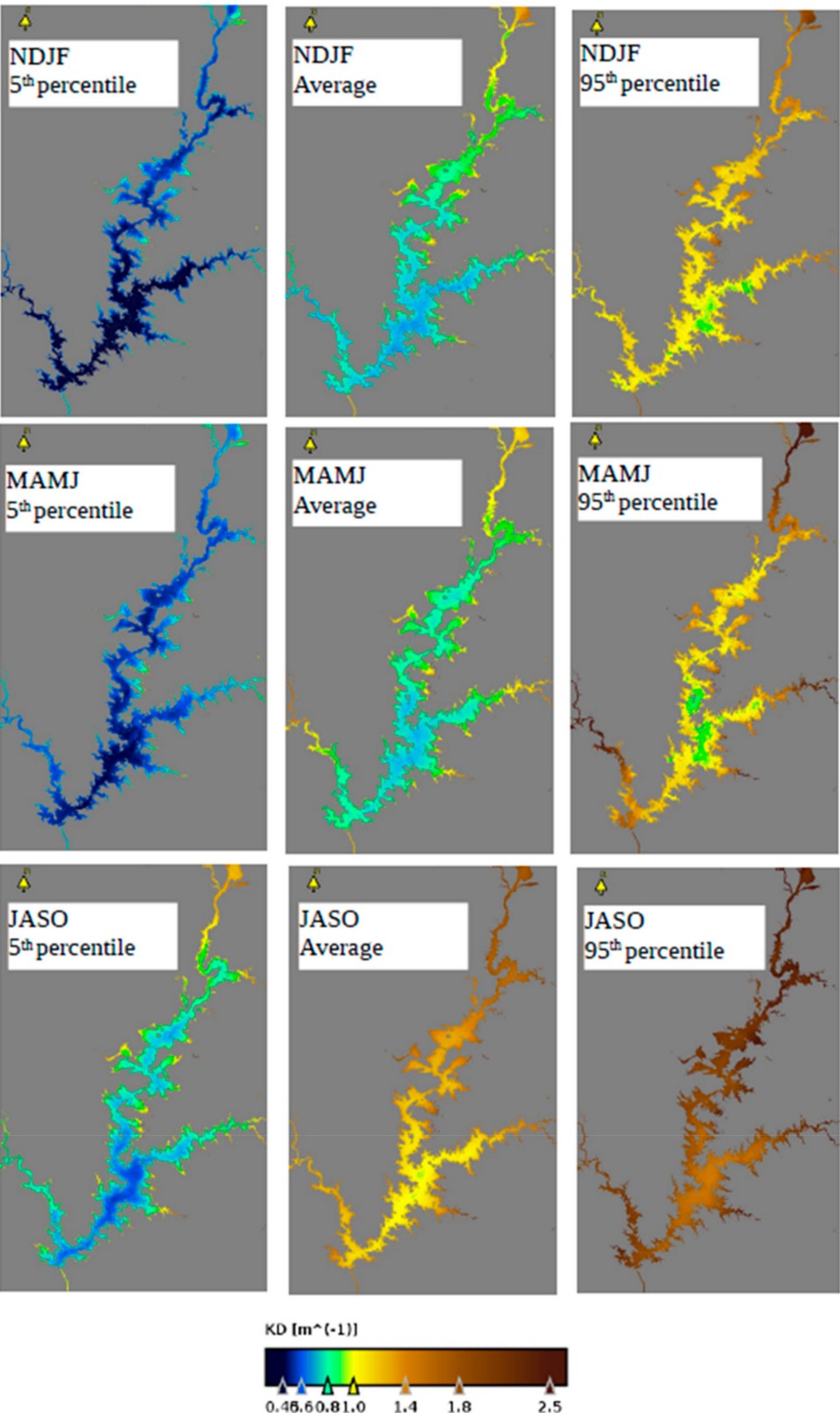

**Figure 9.** Same as in Figure 8 but for the diffuse attenuation coefficient.

As can be seen in Figure 8 and Figure 9, on the days representative of the 5th percentile for KD and 95th percentile for SD there is a greater spatial heterogeneity in the period July–October (JASO), comparatively to the other two analyzed periods. This fact is related to the beginning of the

development of microalgae blooms that usually occur in the northern region of the reservoir mainly by the Guadiana River (Figure 1), usually introducing organic and inorganic matter leading to an increase of biological activity. Only at the end of Summer, these microalgae blooms fill almost the entire reservoir. The southern area of the reservoir, further away from the origin of the microalgae blooms, has the lowest values of KD and the highest SD values. For the other two periods (MAMJ and NDJF), there are also higher SD and lower KD values in the southern region of the reservoir and in larger areas and away from the margins, that are shifting towards the north, with a decrease in SD and an increase in KD. As for the reservoir extensions, to the west and east, they have similar water characteristics to the southern region from November to February, but different patterns in the other periods analyzed. The differences between these regions in the period MAMJ are mainly related to several days of heavy precipitation in spring 2018, as shown in the following section. In the July–October period, more turbidity and lower/higher values of SD/KD respectively were detected in the east and west branches compared to the adjacent southern area, likely to be associated with the shorter distances between margins, with weaker winds in these locations due to less exposure, and thus higher biological activity was expected (Figures 8 and 9—JASO panels). It is observed that in the reservoir in these two years, lower/higher values of SD/KD, respectively, were detected in the July–October period compared to the other two periods, obtaining for almost all reservoir average values of SD lower than 2.5 m and KD over $1 \text{ m}^{-1}$. The maps representing the days with the lowest/highest values of SD/KD respectively (5th percentile to SD and 95th percentile to KD) in the MAMJ and NDJF periods are very similar to the average map in the JASO period, also showing the great deterioration in these months of formation and dissipation of these blooms. Thus, distinct areas in the Alqueva reservoir were selected considering: (1) spatial variation of SD and KD; (2) areas with different widths; (3) geometry, namely the orientation of the lake margins. The selection of the defined areas is presented in five different regions, presented in Figure 10.

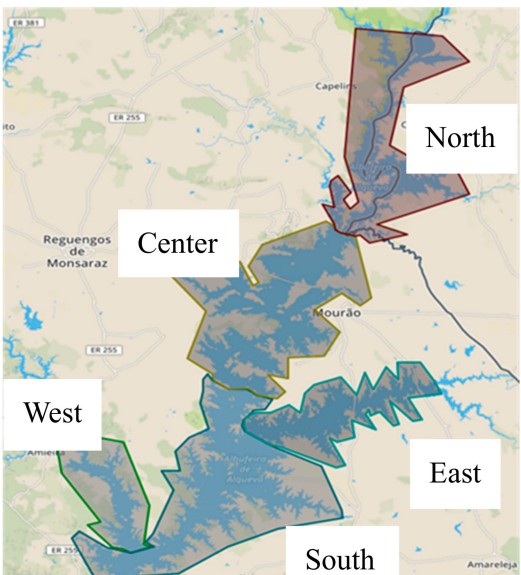

**Figure 10.** Selection of areas according to seasonal variation of water quality parameters, width and geometry of the reservoir.

## 4.4. Spatio-Temporal Variability for Period July 2017 – June 2019

The monthly evolution of SD and KD during the period of July 2017–June 2019, for each selected region of the Alqueva reservoir (Figure 11) was analyzed. When at least one of the regions defined in Section 4.3 less than 50% of water pixels (due to clouds, sunglint or smoke), the satellite image was rejected. To obtain a monthly estimation for each region, the average of all images for each month was

calculated for all water pixels within each area (Figure 11). Water pixels were considered according to the criteria explained in Section 4.3.

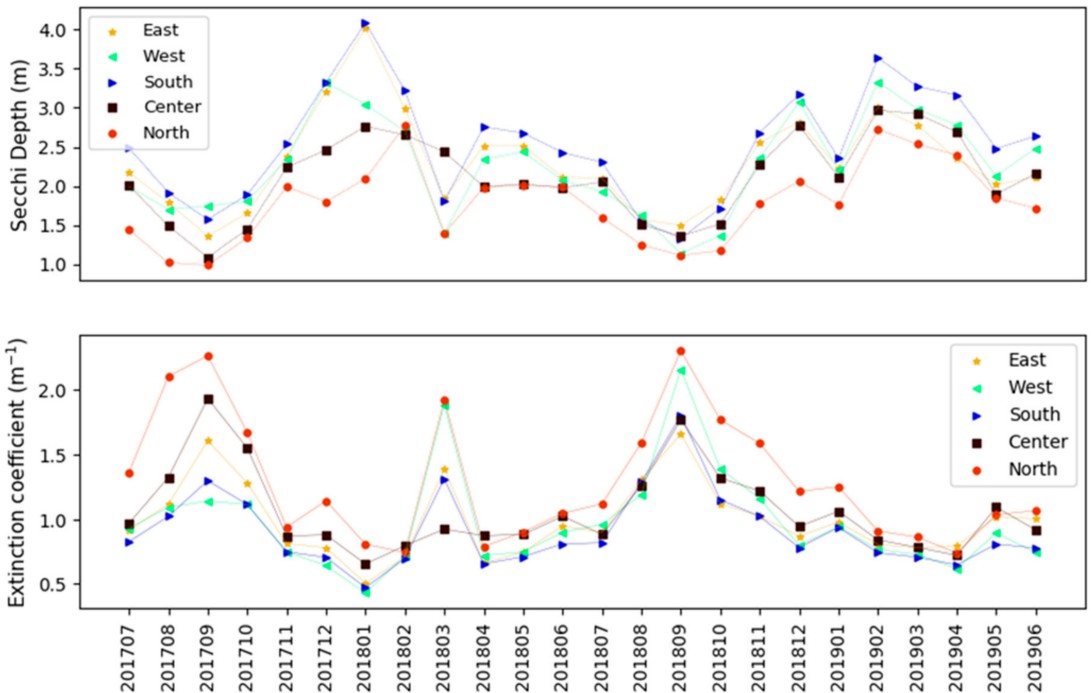

**Figure 11.** Monthly average of SD (top) and KD (bottom) for each reservoir region.

In order to examine conditions that may contribute to changes in water parameters in the reservoir, meteorological data from Cid Almeida station (at the east margin of MontanteP) were used, and compared with estimations of SD and KD, for the 24 months in the period July 2017–June 2019. The meteorological parameters used in this analysis were the monthly wind speed, monthly precipitation, and monthly water and air temperatures. The monthly mean evolution of wind speed, air temperature and water temperature are presented in Figure 12.

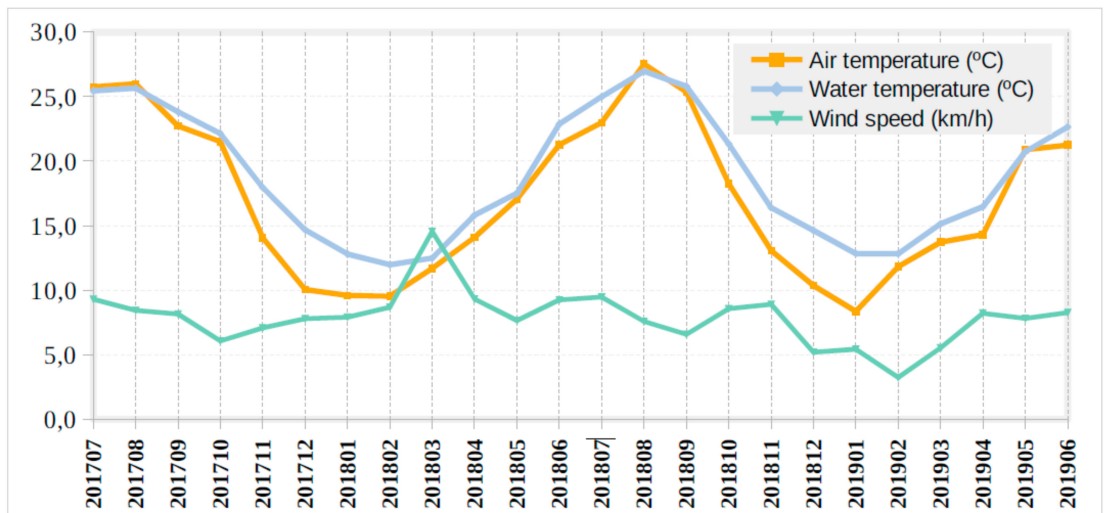

**Figure 12.** Monthly average of water temperature, air temperature and wind speed at the Alqueva reservoir.

In the period July 2017–June 2019, 3 critical periods are identified in terms of SD and KD: August–October (2017), March 2018 and August–October (2018). The variation in the monthly

estimates obtained for each region of SD and KD values is consistent with measurements. The values obtained show higher KD and lower SD values in these periods. Lower KD and higher SD values were estimated in colder months, with lower water temperature, and no significant rainfall events. An excellent relationship between the diffuse attenuation coefficient and turbidity has been identified in several studies (e.g., [60–63]) and also in the Alqueva reservoir with $r^2 = 0.95$ [39]. Since SD is directly related to turbidity, a strong indication of the quality of the estimations of SD and KD for the Alqueva reservoir is that the increase in one of the parameters is related to the decrease in the other.

Typically, in the Alqueva reservoir, critical water quality periods are associated with the usual long and dry summer months, with very high maximum temperatures, typically between 30 °C and 43 °C, inducing the formation of microalgae blooms in the reservoir [64,65]. It is identified that the northern area, between Lucefécit and Monsaraz (Figure 1), presents worse water quality in practically every month, which agrees with that reported by Palma et al. [66] and Palma et al. [67]. Algae blooms in recent years at the Alqueva reservoir developed in the north of Alqueva in July/August, as seen in the example in Figure 9. This is also observed by the estimates of KD and SD in July 2017 and July 2018 (Figure 11), with a significant difference between KD and SD between the northern area and the remaining reservoir regions. As summer progresses, algae expand to the south of the reservoir, with a maximum in September, with this month's SD averages estimations below 1.9 m for all areas, and KD of water over 1.2 m$^{-1}$. The western region of the reservoir, on the other hand, is the area furthest away from the region with the highest presence of algae in the warmer months (North area), however, locally in Summer cases, the development of these microalgae blooms is also possible. For instance, in September 2018, a month with weaker winds and a very hot previous month (August 2018 was the hottest month of the study period), led to the increase in microalgae densities also in this region of the reservoir, with estimations of SD and KD very close to the values of the northern region. This different behavior of the two September's in the evolution of algae on the reservoir and the different estimations of KD/SD can also be seen in the RGB images of the Figure 13, September 19 (2017) without evidence of algae, and September 19 (2018) with great presence of algae in the westernmost area of the reservoir. Except for the West area, the remaining regions of the reservoir behaved similarly in the July–October period for the two years analyzed. During March 2018, SD and KD estimations were like those obtained in the microalgae blooms (August–October), this was due to exceptional rain events. It was recorded nine days with precipitation exceeding 10 mm and almost 200 mm accumulated (Figure 2), a value much higher than average precipitation for March in the Alqueva region. The periods with the highest/lowest SD/KD respectively were December 2017–February 2018 and February–April 2019, periods with low air and water temperatures, associated with anticyclonic weather prevailing on most days, and monthly cumulative rainfall lower than 50 mm, except April 2019. Despite an accumulated rainfall of 70 mm in April 2019, this amount was scattered for 14 days, without heavy precipitation events, with daily precipitation lower than 15 mm. The month which presented the best SD/KD on average was January 2018, with very low KD estimations, of 0.4 m$^{-1}$ for the West area and 0.5 m$^{-1}$ for the South and East area. The best SD estimations were obtained, with average values of 4.0 m and 4.1 m for the East and South areas, respectively. Comparatively, for North and Central regions relevant differences are noticed, with modest SD values of 2.1 m and 2.8 m, respectively. This fact shows that the northern region, between Lucefécit and Monsaraz, presents degraded water quality and may even have large disparities compared with other areas for the same period.

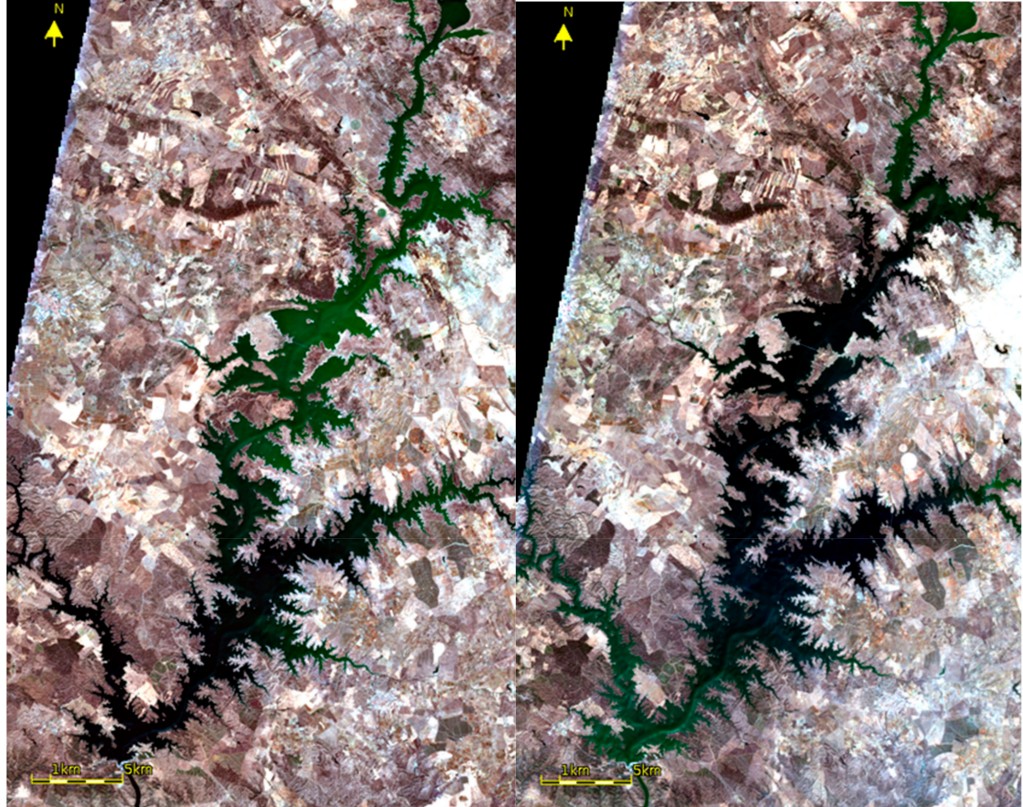

**Figure 13.** RGB Images showing a microalgae bloom for 19 September 2017 (left) and for 19 September 2018 (right) in the Alqueva reservoir.

Alqueva reservoir was classified as eutrophic based on mean total phosphorus (>35 µg P L$^{-1}$) and chlorophyll *a* (>8 µg L$^{-1}$) concentrations [64]. In this study, according to the monthly average of SD in the reservoir, the SD exceeds the limits for the eutrophic status in the northern and central regions of the reservoir for all analyzed months, considering the limit values of the trophic state of lakes and reservoirs [68]. In the Center, West and South regions, some months are already at or very close to the mesotrophic level (SD > 3.0 m), especially in the months with low rainfall and lower lake temperatures.

The southern area of the reservoir presents the best SD/KD values in practically every month analyzed, being expected that two of the main reasons are the widest area between margins (along with the central region) and the longest distance to the North area, where the worst values of the reservoir were found. The central area, between Campinho and Monsaraz, although also with a very large area, presents much highest KD/lowest SD, compared to the southern area for most of the months.

In addition to the monthly average, the 5th and 95th percentiles were calculated to get estimations for pixels representing water with greater and less transparency in each region. This is important to identify possible extreme values in each region and results are presented in Figures 14 and 15.

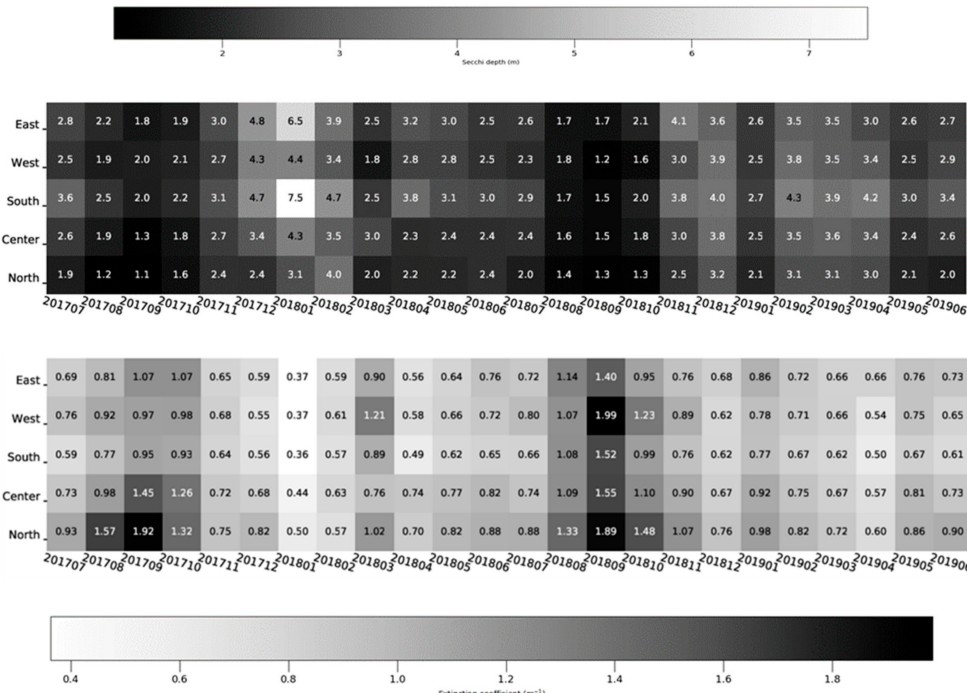

**Figure 14.** The 95th percentile and 5th percentile for SD (top) and for KD (bottom) respectively.

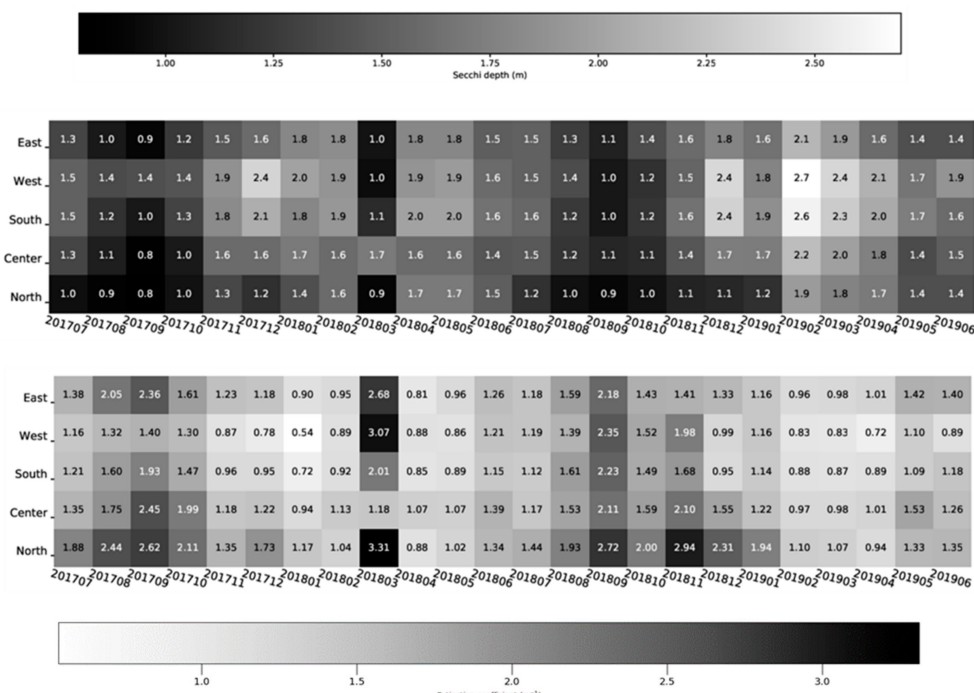

**Figure 15.** The 5th and 95th percentiles for SD (top) and KD (bottom), respectively.

Figure 14 shows the monthly average of pixels representing the highest SD and lowest KD values, and Figure 15 the opposite, that is, less transparent water (lower SD) and with greater radiation attenuation in the water (higher KD).

According to Figure 14, January 2018 features estimates of the highest SD and lowest KD values, with KD below 0.4 m$^{-1}$ in the South, West and East areas, and very high SD values with 6.5 m to the east and 7.5 m to the south. These extreme values of SD and KD are very similar to the in situ extremes measured in the Alqueva reservoir. It should be noted that beyond January 2018 there are other months with a few days presenting SD (95th percentile) > 5.0 m, but which are very smooth on

average for months with large variations in water characteristics. February 2019 presents the smallest spatial variation of KD, that is, representing the smallest spatial variations of dissolved and suspended particles in water, with great uniformity. As an example, note the westernmost area of the reservoir in February 2019, with 90% of the KD values between 0.71 m$^{-1}$ and 0.83 m$^{-1}$. The atmospheric stability in this month, with few precipitation days, and very light winds (Figure 12) should explain this very small spatial variation in the reservoir, characterizing this month as one of the greater transparency periods analyzed between July 2017 and June 2019. Distinctly, November 2018 was the second month in the period analyzed with the highest monthly accumulated precipitation, over 100 mm, and one of the months with the highest average wind, with the largest differences between the 5th percentile and 95th percentile, that is, a greater spatial variation. In the eastern region, SD has a value of 4.1 m (Figure 14) corresponding to 95th percentile (very high value) that is very far from the pixels corresponding to the 5th percentile with low values of only 1.6 m (Figure 15).

It can be seen from Figure 15 that September, in both years analyzed, and March 2018 are the months with the most extreme values of KD and SD. In March 2018, the most extreme KD values were obtained, with a value of 3.31 m$^{-1}$ corresponding to the 95th percentile, e.g., 5% of the pixels in this area with estimated KD of 3.31 m$^{-1}$ or higher. The largest areas (South and West), present for this month the lowest KD values corresponding to 95th percentile and the largest SD corresponding to the 5th percentile, not only on average (Figure 11), as well as for the extreme values. Still, for March 2018, the values for each area show a significant spatial variation in water physical parameters, much higher than for September in both years. This should be explained by the fact that in summer, the spatial variation of algae shows a smaller variation between shores and deeper areas of the reservoir, which is not the case for heavy rainfall, as in March 2018, with much higher values near the margins compared with deeper areas. The slightest differences between the values corresponding to the 5th percentile and 95th percentile, means that the area is homogeneous in water characteristics. For example, in relation to SD in the Northern region, if 90% of the pixels in March 2018 vary spatially between 0.9 m (5th percentile) and 2.0 m (95th percentile), in September 2018, they exhibit great spatial uniformity, with 90% of the pixels for this month presenting estimates between 0.8 m and 1.1 m.

## 5. Conclusions

The methodology developed in this work aimed to characterize spatial and temporal variations of water quality parameters in the Alqueva reservoir, using high spatial resolution images from Sentinel-2 MSI, which can be useful in other lakes with comparable trophic status. A reliable method of atmospheric correction was obtained with the radiative transfer code 6SV, using water vapor and AOT at the reference wavelength of 550 nm, obtained from Sen2Cor. Empirical relationships between satellite retrieved surface spectral reflectances and KD/SD measurements were obtained and subsequently validated, obtaining high correlation coefficients and low NRMSE values below 20% for the two analytically defined algorithms. In general, the southern area of the reservoir (between Montante e Campinho) features the highest SD and lowest KD values in the analyzed period. The western (Between Montante and Álamos) and eastern sections (Alcarrache) of the reservoir presented slightly lowest SD and highest KD values, compared to the southern part of the reservoir, especially during or immediately after extreme precipitation events, and in the August–October period. Every month, the northern region, between Lucefecit and Monsaraz, presents more turbid water and with greater attenuation of radiation in the water. As for the temporal variations, the Alqueva reservoir presented the lowest SD and highest KD values in August–October period and March 2018 compared to the other periods of the year. Between August and October, there is the usual formation of microalgae blooms in the reservoir, inducing average estimated SD between 1.9 m in Central area and 1.0 m to North area (September of 2017). The mean values of KD also denote more attenuation of radiation in water in this time of the year, with values between 1.02 m$^{-1}$ in the southernmost area of the reservoir and 2.27 m$^{-1}$ in the North area.

With exception of months with excessive precipitation and with the presence of microalgae, higher values of SD and low KD were obtained noting the SD of 7.5 m for pixels representing 95$^{th}$ percentile and 0.36 m of KD for pixels representing 5th percentile in South region to January 2018. These extremes obtained from the satellite are very close to the extremes measured in the field. A strong indicator of the correctness of the methodology presented is the good agreement between the evolution of SD estimations and KD with the evolution of meteorological parameters/water temperature. The highest KD/lowest SD was observed in the highest water temperature/March 2018 (very rainy month), with a smaller spatial variation of these parameters for months of low wind speed.

Highest KD and lowest SD were observed in the warmest months/highest water temperature, and in March 2018, they presented monthly accumulated precipitation of around 200 mm. On the other hand, estimations of the highest SD and lowest KD values were obtained in the months of lower monthly accumulated precipitation, coupled with lower air and water temperatures. A smaller spatial variation of these parameters is presented, for months of low wind speed, as is a clear example of February 2019, the month with lower monthly wind intensities. The months with the highest monthly rainfall, such as March 2018 and November 2018, show much greater spatial variations of SD and KD due to the large difference in values between margins compared to deeper zones.

The observations show high spatial and temporal heterogeneities in SD and KD. This means that the use of spatial and temporal constant values in NWP models (as at present) may lead to errors in the surface energy budget over water and subsequently in surface temperature forecasts. The methodology and the empirical relationships found in this work show that it will be possible to get global maps from satellite remote sensing for use in NWP models.

**Author Contributions:** Conceptualization, G.R., M.P., M.J.C.; methodology, G.R., M.P., M.J.C.; software, G.R.; validation, G.R., M.P., M.J.C., M.H.N., A.M.P., R.S., M.M.M.; formal analysis, G.R., M.P., M.J.C.; investigation, G.R., M.P., M.J.C.; resources, G.R., M.P., M.J.C.; data curation, G.R., M.P., M.J.C.; writing—original draft preparation, G.R.; writing—review and editing, G.R., M.P., M.J.C., M.H.N., A.M.P., R.S., M.M.M.; visualization, G.R.; supervision, M.P., M.J.C.; project administration, R.S. All authors have read and agreed to the published version of the manuscript.

**Funding:** The work was supported by COMPETE 2020 through the ICT project (UIDB/04683/2020) with the reference POCI-01-0145-FEDER-007690 and through the ALOP project (ALT20-03-0145-FEDER-000004).

**Acknowledgments:** The authors thank Copernicus and ESA for the data Sentinel-2 provided to carry out this work and to Olivier Hagolle for providing open-source codes to massive Sentinel-2 data download.

**Conflicts of Interest:** The authors declare no conflict of interest.

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
