# Peer review of "Temporal and Spatial Variations of Secchi Depth and Diffuse Attenuation Coefficient from Sentinel-2 MSI over a Large Reservoir"

_remotesensing, doi:10.3390/rs12050768_

Round 1
Reviewer 1 Report
This manuscript (1) presents methods for estimating Secchi Depth and the extinction coefficient using Sentinel-2 MSI, (2) applies the developed methods to MSI data covering a large reservoir to study the spatial and temporal variability. It is important to have field measurements to validate ocean colour satellite data, hence this study is valuable for using satellites to observe the water quality of the Alqueva reservoir. However, in its current form, the manuscript is sometimes unclear and ambiguous, particularly in the methods sections, and some of the figures need more attention – see below for more detailed comments.
General comments/questions:
The authors use the term “extinction coefficient” without fully defining it. They give it the symbol KD and refer to their previous publications. From this, I believe when the authors state “extinction coefficient” they mean the “diffuse attenuation coefficient” – the term which is most commonly used in ocean colour remote sensing. I recommend the authors use this language to increase the readability and understanding of their manuscript. That being said, I think the authors are discussing the diffuse attenuation coefficient of radiance (not irradiance), which typically is given the symbol K, not K_d (see Mobley, 1994, Light and Water or the Ocean Optics Web Book).
Further, the authors don’t explicitly describe the method used to measure KD, they state it was “made using the method described in Potes et al [20]”. I think it would be useful to provide a brief overview of the method used to derive KD in this manuscript. It would help to clarify what the authors mean by “extinction coefficient” or even “diffuse attenuation coefficient”. Additionally, there are a few different ways of deriving KD from a light profile, so it would be useful to know exactly what the authors did e.g. did you use the slope of a line fitted through the logarithm of the light profile and depth? Did you use all depths or discard some?
It would be useful to explain somewhere the motivation for evaluating the two atmospheric correction methods. Additionally, in the final paragraph of the introduction, I suggest stating one of your objectives is to also test two atmospheric correction methods. At the moment, the way the manuscript is written, I was surprised when I reached section 3.1 and the corresponding results because I wasn’t expecting them at all.
I understand with high spatial resolution satellites we lose the temporal resolution, thus the authors selected overpasses within a day of in situ measurements. The recommended time difference for algorithm development is 3 hours (Bailey and Werdell, 2006). I suspect if the authors used that time window they would have very few matchups, hence selecting the 1 day window they use. Could the authors please comment on this explicitly in the text?
The authors state “common statistical indicators have been used in this study”. However, it is not explicitly clear exactly what they did. Providing equations would be useful. Additionally, later in the manuscript, the authors use the term NRMSE without defining it. I also recommend Seegers et al (2018) who provide a good overview of performance metrics to use when assessing satellite data products.
It isn’t clear how the authors chose what empirical relationships to test for the SD and KD algorithms. Did you test relationships from the literature? Did you try every combination of reflectance bands? Why didn’t you use any relationships from the literature? There are lots of KD and SD algorithms out there and the authors don’t mention them at all.
When it comes to the validation of the KD and SD algorithms, the authors don’t quote the bias and the slope of the SD validation result is quite far from 1.
The description at the start of section 4.3 about identifying water pixels isn’t explicitly clear. How were the “errors corrected” (line 392)? In Section 5 (Conclusions, line 581) the authors mention using “water vapour and AOT”, but that isn’t mentioned anywhere in section 4.3.
Towards the end of the manuscript, the authors use the terms “best” and “poorest” a lot to describe the SD and KD values. What do they mean by “best” and “poorest”? They are not scientific terms. I suggest using “lowest” and “highest” accordingly.
Comments on Figures and Tables:
Suggest switching the order of figure 1 and figure 2. Figure 2 is mentioned first in the text and, as the map of the region, is the more logical one to go first.
Fig 3: What are the units? I’m not familiar with “dl”, reflectance is typically given in sr^-1
Fig 3: What are the lines? I’m assuming the dashed is one-to-one, and solid are best-fit lines of some sort. Please can you provide those details in the text/figure caption
Table 3: what are the units of Bias and RMSE? Please provide those in the table
I suggest including a Figure showing the data that were used to derive the SD and KD relationships e.g. B2/(B3+B4) vs SD. You don’t mention how many data points were used to derive the relationships and a visual representation of the algorithm would be useful.
Fig 4 & 5: There are no ticks on the axes, so it is very difficult to read the values of the data points.
Figure 6: Why is the land so different coloured? Have the RGB channels been scaled differently? If so, then the colour of the water is not directly comparable.
Figure 10: Suggest labelling each of the areas.
Specific comments:
L18 “it is presented” should be “we present” or “a methodology developed to … is presented”
L21: “Empiric” should be “empirical”
L25: “which allowed differentiating” should be “which allowed us to differentiate” or “which allowed five zones to be differentiated”
L77: suggest replacing “achieve” with “derive”
L82: suggest rewording to “Landsat also has a high spatial resolution (30 m), but a much …”
L84: should be “MODIS has” not “MODIS have”
L112: MontanteP or MontanteA?
L139-149: Perhaps move to the results section?
L155: “acquired in” not “acquired on”
L155: suggest adding the word “spatial” i.e. “very high spatial resolution”
L156: remove “of radiometric resolution”
L174: replace “yet” with “which is”
L175: replace “correction” with “removal”
L177: replace “regard” with “are”
References:
Mobley (1994) Light and Water
Bailey and Werdell (2006) A multi-sensor approach for the on-orbit validation of ocean color satellite data products. Remote Sens of Environ, 102, pp12-23
Seegers et al (2018) Performance metrics for the assessment of satellite data products: an ocean color case study. Optics Express, 26 (6), 7404-7422
Author Response
Dear Reviewer,
Attached you can find our comments

Reviewer 2 Report
Please check the attached file.

Author Response
Dear Reviewer,
We send our comments attached.

Reviewer 3 Report
1.An interesting paper need to be slightly improved.
2.Stat of the art review must be updated. Provide detail about the studies conducted worldwide and related to modelling SD using several kinds of models, especially machines learning models.
3.Results and analysis need to be improved. Using only RMSE, MAPE and bias (table 3) is insufficient. The NSE, d and R2 are useful indices.
4.Quality of figures needs to be improved. For example figure5 is unclear, the R2 is missing.
5.Results must be presented separately in the training and validation phases as reported in in (lines 298-300).
Author Response

(The authors gave the same response as above.)

Round 2
Reviewer 1 Report
The authors have responded to most of comments adequately. However, I have a few more comments - see the attached document.

Author Response
The authors thank the reviewer for the valuable comments that greatly contributed to the improvement of the manuscript. Our answers are attached.

Reviewer 2 Report
The author have carefully modified the paper according to my all comments. Now it could be accepted.
Author Response
The authors thank the reviewer for the valuable comments that greatly contributed to the improvement of the manuscript.